https://doi.org/10.1038/s41467-020-14706-1　　**OPEN**

# Optoacoustic brain stimulation at submillimeter spatial precision

Ying Jiang[1,2], Hyeon Jeong Lee [2,3,4,5,6], Lu Lan[2,5], Hua-an Tseng[5], Chen Yang[2,4], Heng-Ye Man [1,7], Xue Han [1,2,5] & Ji-Xin Cheng [1,2,4,5 ✉]

Low-intensity ultrasound is an emerging modality for neuromodulation. Yet, transcranial neuromodulation using low-frequency piezo-based transducers offers poor spatial confinement of excitation volume, often bigger than a few millimeters in diameter. In addition, the bulky size limits their implementation in a wearable setting and prevents integration with other experimental modalities. Here, we report spatially confined optoacoustic neural stimulation through a miniaturized Fiber-Optoacoustic Converter (FOC). The FOC has a diameter of 600 μm and generates omnidirectional ultrasound wave locally at the fiber tip through the optoacoustic effect. We show that the acoustic wave generated by FOC can directly activate individual cultured neurons and generate intracellular $Ca^{2+}$ transients. The FOC activates neurons within a radius of 500 μm around the fiber tip, delivering superior spatial resolution over conventional piezo-based low-frequency transducers. Finally, we demonstrate direct and spatially confined neural stimulation of mouse brain and modulation of motor activity in vivo.

---

[1] Graduate Program for Neuroscience, Boston University, Boston, MA 02215, USA. [2] Photonics Center, Boston University, Boston, MA 02215, USA. [3] College of Biomedical Engineering and Instrument Science, Zhejiang University, Hangzhou, Zhejiang 310027, People's Republic of China. [4] Department of Electrical and Computer Engineering, Boston University, Boston, MA 02215, USA. [5] Department of Biomedical Engineering, Boston University, Boston, MA 02215, USA. [6] Provincial Key Laboratory of Cardio-Cerebral Vascular Detection Technology and Medicinal Effectiveness Appraisal, Zhejiang University, Hangzhou 310027, People's Republic of China. [7] Department of Biology Sciences, Boston University, Boston, MA 02215, USA. ✉email: jxcheng@bu.edu

Ultrasound is an emerging neuromodulation technique that offers the potential of non-invasively modulating brain activities[1,2]. Early reports of neuromodulation using high-intensity ultrasound date back to the 1920s, likely through a tissue heating mechanism[3,4]. In the past decade, neural stimulation using low-intensity, low-frequency focused ultrasound has been demonstrated to directly evoke action potentials and modulate motor responses in rodents[2,5], rabbits[6], non-human primates[7], and sensory/motor responses in humans[8–11] through a non-thermal mechanism. Two recent papers by the Shapiro group[12] and Lim group[13] argued that these responses could be a consequence of indirect auditory stimulation through the cochlear pathway. On the other hand, Tyler, Baccus, Shoham, Pauly, and their coworkers reported direct activation of neurons in brain slices[14], isolated retina[15,16], and deaf mice[17], where no auditory circuitry is involved. A major challenge facing ultrasound neural modulation, which contributes to the mentioned controversies, is that delivery of transcranial ultrasound would inevitably go through the skull, and eventually reach the cochlea through bone transduction. Moreover, the presence of the skull will reflect acoustic waves and compromise the ultrasound focus, resulting in a spatial resolution of a few millimeters, which is insufficient for region-specific brain stimulation in small rodents.

An alternative way to generate ultrasound wave is through the optoacoustic effect. In an optoacoustic process, pulsed light is illuminated on an absorber, causing transient heating and thermal expansion, and generating broadband acoustic waves at ultrasonic frequencies[18]. Recently, the optoacoustic effect has received increasing attention in the fields of imaging and translational medicine[19,20]. Using endogenous, as well as exogenous absorbers, optoacoustic tomography, and microscopy have found broad biomedical applications[21,22]. More specifically, optoacoustic tomography has allowed imaging of brain structures, as well as function in a non-invasive manner[23–26]. Beyond imaging, recent advances in developing optoacoustic materials have enabled highly efficient optoacoustic conversion[27]. Pulsed light excitation of these optoacoustic materials generates ultrasound waves at high amplitude, which allowed for all-optical ultrasound imaging[28,29], tissue cavitation[30,31], and precision surgical guidance of lumpectomy[32].

Here, we report the use of acoustic waves generated by the optoacoustic process for direct and spatially confined neural stimulation both in culture and in vivo in a functional brain. The stimulation is based on a novel fiber-optoacoustic converter (FOC) that generates omnidirectional low-frequency ultrasound pulses emitting from a coated fiber tip. The miniaturized size of the FOC together with a rapid attenuation of optoacoustic intensity with distance provides superior spatial confinement of the generated ultrasound. By time-resolved calcium imaging, we demonstrate that the FOC can reliably produce neural activation within a 500 μm radius from the FOC tip in cultured neurons. By combining FOC with in vivo electrophysiology recordings, we achieved direct optoacoustic activation of mouse somatosensory cortex in the mouse brain, providing evidence that the observed activation is a consequence of direct neural stimulation without the involvement of the cochlear pathway. Finally, we demonstrate functional modulation of the motor cortex by showing that FOC can evoke motor responses with high-spatial precision.

## Results

### Fabrication and characterization of FOC.
The FOC is composed of a passively Q-switched diode-pumped solid-state laser (PQSY, RPMC USA) centered at 1030 nm with a pulse width of 3 ns and pulse energy of 100 μJ, a 200-μm core diameter, ~0.5 m long, 0.22 NA multimodal fiber, and a ball-shaped coated tip with a diameter of ~600 μm (Fig. 1a). Through the optoacoustic process, the pulsed laser energy is converted into acoustic waves generated at the FOC tip. The acoustic waves then excite neurons in the proximity to the tip. The FOC tip was coated with a 2-layer nano-composite (Fig. 1b). The first layer is a diffusion layer composed of a mixture of ZnO nanoparticles in epoxy (15% w/w). The ZnO nanoparticles have a 100-nm diameter, which is smaller than the wavelength of the incident light and enables Raleigh scattering of the light. Consequently, the incident light is randomly scattered in all directions, which produces a relatively uniform angular distribution of the laser pulse (Supplementary Fig. 2). The second layer is an absorption layer composed of a mixture of graphite powders in epoxy (30% w/w). With its high optical absorption and thermal conduction efficiency, the graphite completely absorbs the diffused laser and converts it into heat. The heat is then transferred to surrounding epoxy, creating expansion and compression of the epoxy, and generating acoustic waves that propagate in an omnidirectional manner. To characterize the generated acoustic wave from FOC, we applied the nanosecond laser at a pulse energy of 14.5 μJ, and measured the acoustic wave by an ultrasound transducer underwater. A representative acoustic wave generated by a single laser pulse is shown in Fig. 1c. The radiofrequency spectrum shows that the generated acoustic wave is in the ultrasound frequency ranging from 0.5 to 5 MHz, with multiple peaks between 1 and 5 MHz (Fig. 1d). The maximum acoustic pressure is measured to be 0.48 MPa using a needle hydrophone. To examine the angular distribution of the acoustic wave, we measured the pressures at various angles. The distribution map shows that the intensity is the strongest in the forward direction, while the back-propagating ultrasound is about 50% of the forward intensity (Fig. 1e). We note that fiber-based optoacoustic generation with a single layer of absorber has been reported[33,34]. A novel component of our FOC is the double-layer structure, where the diffuse layer not only results in an omni-directional acoustic wave, but also significantly lowers the acoustic wave frequency (Supplementary Fig. 1). Low-frequency ultrasound was shown to be more efficient for neural modulation[35].

### FOC stimulation in vitro with high-spatial precision.
To investigate whether the FOC can directly modulate neuronal activity, we examined the response of cultured neurons to FOC stimulation. We treated rat cortical neurons (days in vitro 18 to 22) with a calcium indicator, Oregon Green™ 488 BAPTA-1 dextran (OGD-1), and performed calcium imaging (Fig. 2a) using an inverted wide-field fluorescence microscope (Supplementary Fig. 3). The FOC was placed ~100 μm above the focal plane, in the center of the field of view. The acoustic wave was produced with laser pulses at a repetition rate of 3.6 kHz over a period of 200 ms, which corresponds to ~720 pulses (Supplementary Fig. S4). Calcium transients were observed for all neurons in the field of view (max $\Delta F/F = 9.5 \pm 2.9\%$, $n = 36$ cells from three cultures, data in mean ± SD) (Fig. 2b). Addition of 15 μM intracellular calcium chelator BAPTA-AM significantly reduced the calcium signal (max $\Delta F/F = 2.6 \pm 0.5\%$, $n = 12$,) (Supplementary Fig. 5) compared to neurons without BAPTA-AM ($p = 1.17 \times 10^{-5}$, two-sample $t$-test). The response latency was found to be <50 ms, since the responses were observed at the first frame post stimulation onset across experiments with a camera acquisition rate at 20 Hz. To identify the threshold for FOC-induced neural activation, we varied the stimulation duration to 100, 50, and 20 ms. The FOC successfully produced neural activation with 100 and 50 ms stimulation, but not with 20 ms stimulation (Fig. 2c). We next asked whether the FOC can produce neural activation reliably and repeatedly. Eight bursts of laser pulses, each with 200 ms

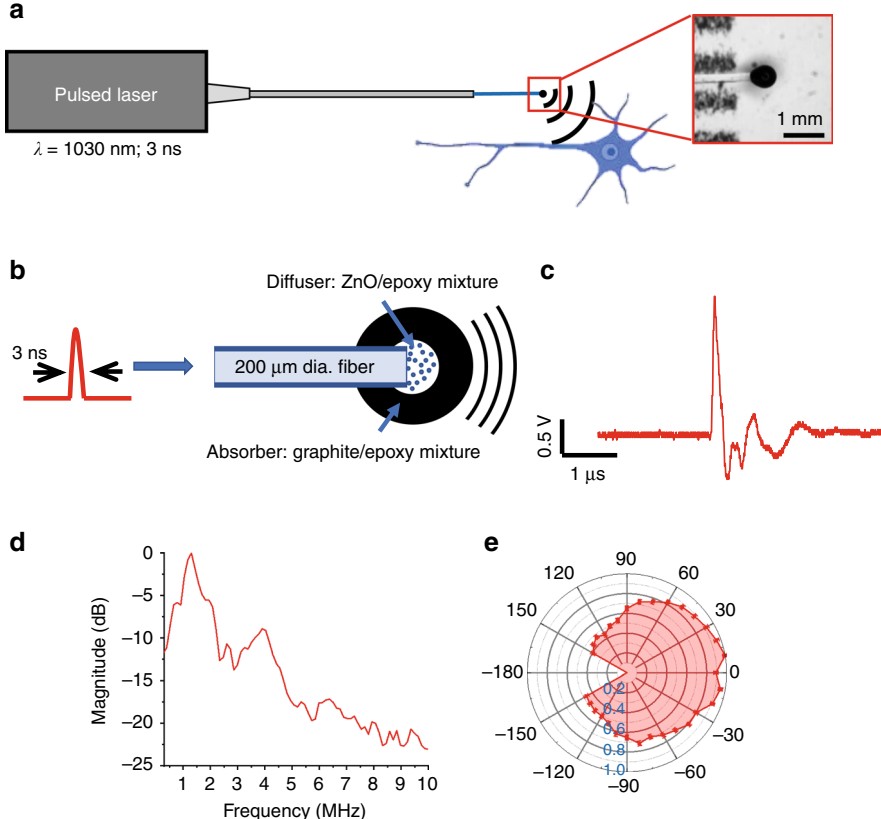

**Fig. 1 Design of FOC and characterization of the FOC-generated acoustic wave. a** The concept of optoacoustic neuromodulation through a FOC. Insert is an enlarged FOC tip under a stereoscope. **b** Schematic of acoustic wave generation. **c** Representative acoustic wave recorded with a transducer. **d**, **e** radiofrequency spectrum and angler intensity distribution of FOC-generated acoustic wave. Error bar: ± SD (1 fiber, 3 repeats) Source data are provided as a Source Data file.

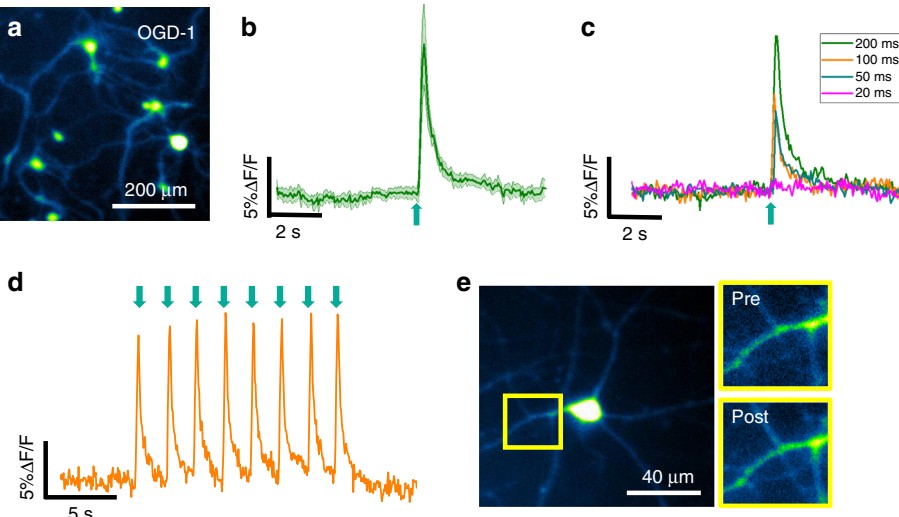

**Fig. 2 FOC induces calcium transients in cultured primary neurons. a** Neurons loaded with OGD-1. **b** The average trace of neuronal calcium trace ($n = 12$) in response to 200 ms FOC stimulation. Shaded area: ± SD. **c** Representative traces of neuronal response to 200-ms, 100-ms, 50-ms, and 20-ms FOC stimulation. **d** Calcium trace of a neuron undergone repeated FOC stimulations. **e** Representative image of a neuron before and after repeated FOC stimulations. Green arrows: stimulation onset. Source data are provided as a Source Data file.

burst duration and 2-s inter-burst interval, were delivered to the FOC. Stable calcium transients in response to each laser pulse train were observed (max $\Delta F/F = 12 \pm 0.6\%$, eight pulses) (Fig. 2d) (Supplementary Movie 1). No obvious morphological changes were detected in neurons stimulated multiple times over a 2-h duration (Fig. 2e).

To verify that the FOC-induced calcium transients are specific to neurons, we loaded OGD-1 to a prostate cancer cell line (PC3).

By 200-ms FOC stimulation, we observed a calcium signal (max $\Delta F/F = 0.7 \pm 2.7\%$, $n = 52$) significantly smaller than that in neurons ($p = 1.98 \times 10^{-7}$, two-sample $t$-test) (Supplementary Fig. 6). Next, we obtained a rat glial culture loaded with OGD-1 and delivered 200-ms FOC stimulation to morphologically identified astrocytes, and observed responses (max $\Delta F/F = 1.2 \pm 0.6\%$, $n = 82$) significantly smaller than the calcium transients produced by FOC stimulated neurons ($p = 1.16 \times 10^{-6}$, two-sample $t$-test) (Supplementary Fig. 7). The glial culture was immuno-stained after experiments and was confirmed as GFAP positive. These data suggest that the FOC reliably and selectively activates neuronal cells in vitro.

A key advantage of FOC over traditional ultrasound transducers is that the FOC emits pulsed ultrasound waves locally at the coated fiber tip, which allows localized stimulation. If homogeneous omnidirectional propagation is assumed, as the wavefront propagates and expands, the acoustic intensity $I$ at distance $d$ can be described as $I_d = I_0 \cdot \left(\frac{r}{r+d}\right)^2$, where $I_0$ is the intensity at the FOC surface, and $r$ is the radius of the FOC tip. Since the acoustic pressure $p$ is $p = \sqrt{I \cdot Z}$, where $Z$ is the acoustic impedance. This gives the acoustic pressure $p$ at distance $d$ as $p_d = \frac{r}{r+d}\sqrt{I_0 \cdot Z}$. Thus, the acoustic intensity is expected to attenuate quickly when propagating in the medium. To experimentally characterize the attenuation of acoustic wave with distance, we used a transducer array to perform wavefront reconstruction. Wavefronts at 6 different time delays with 0.4 µs interval were stitched together in Fig. 3a. We fitted the curve with the equation described above. With a FOC tip of 600 µm in diameter, the acoustic intensity is attenuated by 61% at 1.0 mm away from the tip underwater (Fig. 3b). This result demonstrates submillimeter spatial precision. Since the brain tissue has a much higher ultrasound attenuation coefficient (0.6 dB/cm MHz) than water (0.0022 dB/cm MHz)[36], we expect the FOC to produce even more localized acoustic wave inside the brain. To demonstrate that FOC-induced neural activation is spatially confined, we placed the FOC at the edge of the imaging field of view and delivered a laser pulse train of 200 ms duration (Fig. 3c). It was observed that neurons within 500 µm distance from the FOC showed reliable calcium response, and that the amplitude of the response is highly dependent on the relative distance from the FOC (Fig. 3d). When we sorted the neurons by their distance from the FOC, we found that neurons that are within 500 µm to the FOC showed strong responses (max $\Delta F/F = 9.4 \pm 3.4\%$, $n = 10$), while neurons that are 500 µm to 1.0 millimeter away showed significantly smaller responses (max $\Delta F/F = 1.5 \pm 1.0\%$, $n = 13$, $p = 1.9 \times 10^{-4}$, two-sample $t$-test) (Fig. 3e). These data demonstrate that the effect of FOC is highly localized within a 500 µm radius, which provides one order of magnitude better spatial resolution comparing to the widely used piezo transducers having a several-millimeter focus area.

To eliminate the possibility that the activation is due to laser illumination, we measured the leaked light energy from the FOC tip with a photodiode and found only 0.11% of the laser leaked out of the FOC. Additionally, we used an uncoated optical fiber and delivered the same laser pulses with 10% of the laser intensity at 3.6 kHz repetition rate and 200 ms duration directly to the neuron. No calcium transients were observed (Supplementary Fig. 8a). To examine the possibility of photothermal neural activation, we measured the heat profile of the FOC tip using a miniaturized ultrafast thermal probe. The temperature increase on the FOC surface was found to be 1.6, 0.9, 0.5 °C for 200, 100, 50 ms laser stimulations, respectively (Supplementary Fig. 8b). Such temperature increase is well below the previously reported threshold for thermal induced neural activation ($\Delta T > 5$ °C)[37]. Therefore, the neural activation effect is most likely contributed by the generated acoustic wave.

**FOC induces direct activation in mouse brain**. Since we demonstrated that the FOC can reliably stimulate neurons with high-spatial precision in vitro, we next moved on to investigate whether FOC can successfully induce neural activation in vivo in the mouse brain, with similar spatial precision. A mouse was anesthetized with 1% isoflurane, and a cranial window was made

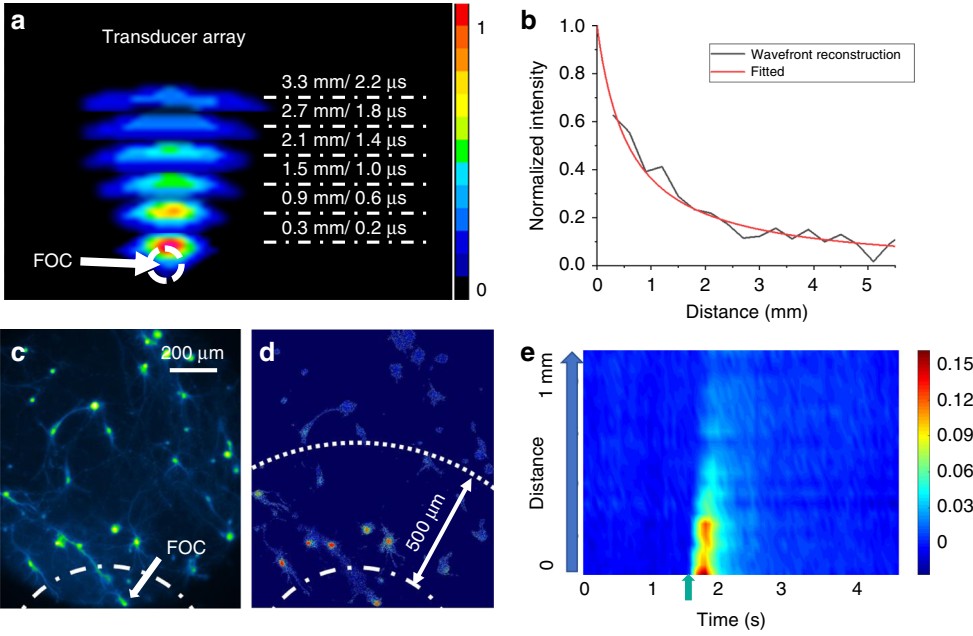

**Fig. 3 Spatially confined acoustic wave and neural stimulation induced by FOC. a** FOC wavefront reconstruction by the transducer array. Note only part of the wavefront is reconstructed due to the limited receptive angle of the transducer array. **b** Acoustic intensity attenuates significantly as the distance to the FOC increases. **c, d** Spatial distribution of maximum neuronal calcium response induced by 200 ms FOC stimulation. Dashed line: placement of FOC. **e** Sorted calcium traces of neurons by the distance from the cell to the FOC. Green arrow: stimulation onset. Source data are provided as a Source Data file.

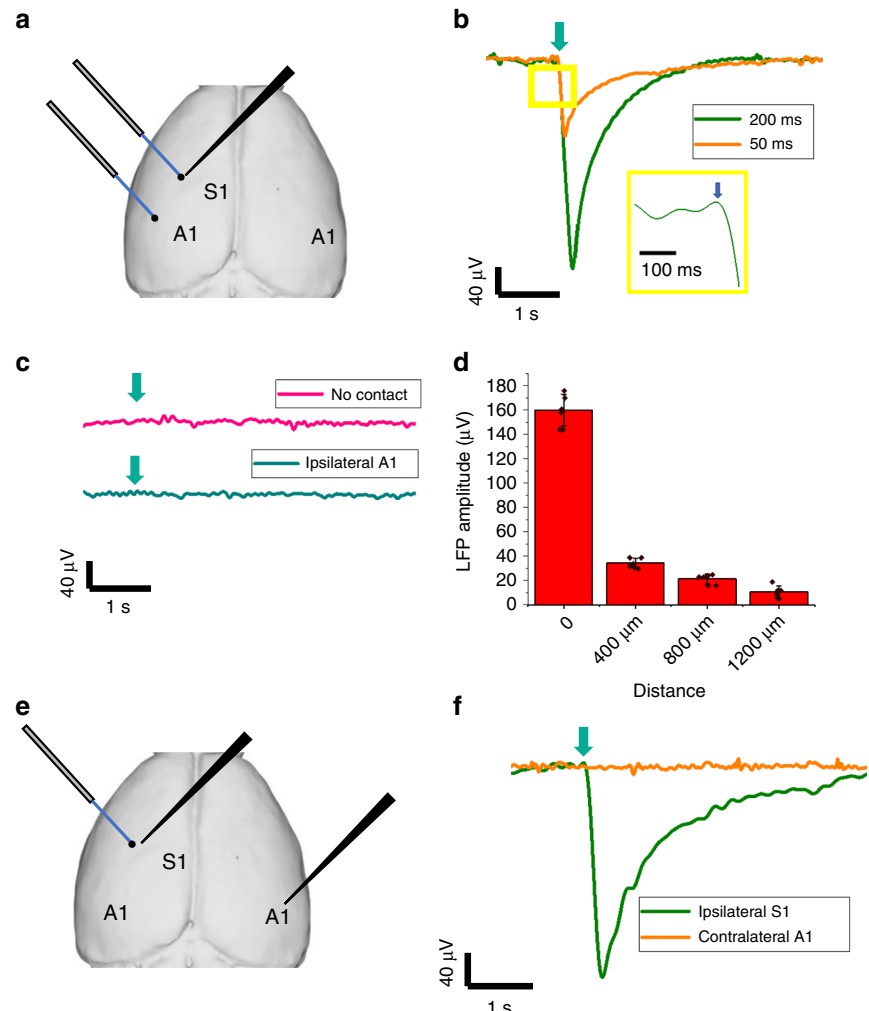

**Fig. 4 FOC induces neural activation in vivo in the mouse brain. a** Placement of FOC in S1 and A1, and ipsilateral recording electrode in S1 test the spatial confinement of FOC stimulation. **b** S1 LFP response to 50 ms and 200 ms FOC stimulation delivered to S1. Insert: zoomed in LFP trace showing response latency. **c** S1 LFP response to S1 LFP response to FOC and stimulation delivered to S1 without contact FOC stimulation delivered to ipsilateral A1. **d** LFP amplitude recorded at different distances from the FOC. **e** Placement of FOC in S1 and recording electrode in ipsilateral S1 and contralateral A1 to test the involvement of the auditory pathway. **f** LFP response pf ipsilateral S1 and contralateral A1 to S1 FOC stimulation. Error bar ± SD. Green arrow: stimulation onset. Source data are provided as a Source Data file.

above the primary somatosensory cortex (S1) and primary auditory cortex (A1) based on stereotaxic coordinates with the dura intact (Fig. 4a). First, we investigated whether FOC can produce activation of the local cortical area. The FOC was brought close to contact the brain surface and immersed in saline. Laser pulse trains with 200 ms and 50 ms duration were delivered to the FOC, and neural activities were recorded with a tungsten electrode. We observed robust local-field potential (LFP) response to the FOC stimulation for both stimulation durations, with response latency of $15.87 \pm 1.34$ ms ($n = 9$, from three mice) (Fig. 4b), which is indicative of direct neural activation. When the FOC was lifted up by 100 μm without contacting the brain or immersing saline, the FOC failed to induce any neural activation (Fig. 4c). This result indicates that the neural activation is induced by acoustic waves that have minimal propagation in the air. Next, we delivered FOC stimulation to the ipsilateral A1, which is ~2 mm away from the S1 recording site. No neural response was detected, which indicates that the FOC stimulation was spatially confined to S1 (Fig. 4c). To further quantify the spatial precision in vivo, we recorded the LFP responses as the FOC moves away from the recording electrode. The LFP amplitude drops from

$159.8 \pm 13.2$ μV to $10.5 \pm 5.1$ μV at 400 μm away ($n = 3$) (Fig. 4d), demonstrating superior spatial confinement of FOC stimulation in vivo.

Since ultrasound stimulation by auditory pathway has been reported[12,13], we examined whether the auditory pathway is involved in the FOC stimulation. One cranial window was made above the S1 region, and another on the contralateral A1 region. The FOC stimulation was delivered to the S1, and the recording electrode was placed in the ipsilateral S1 or contralateral A1 (Fig. 4e). If the auditory pathway is involved, we would expect to observe strong responses in the contralateral A1 with ~50 ms delay[13]. However, a FOC stimulation of 200 ms duration on the S1 evoked robust LFP response on ipsilateral S1, but failed to evoke any response in the contralateral A1 (Fig. 4f). Finally, to rule out the possibility of laser or ultrasound-induced electrical artifact of the electrode, we record voltage change on the FOC surface in saline and found that laser pulses of 200 ms duration produced no voltage change on the FOC tip (Supplementary Fig. 9). Collectively, these data suggest that the FOC produces direct neural stimulation in vivo with high-spatial and temporal precision, without the involvement of the auditory pathway.

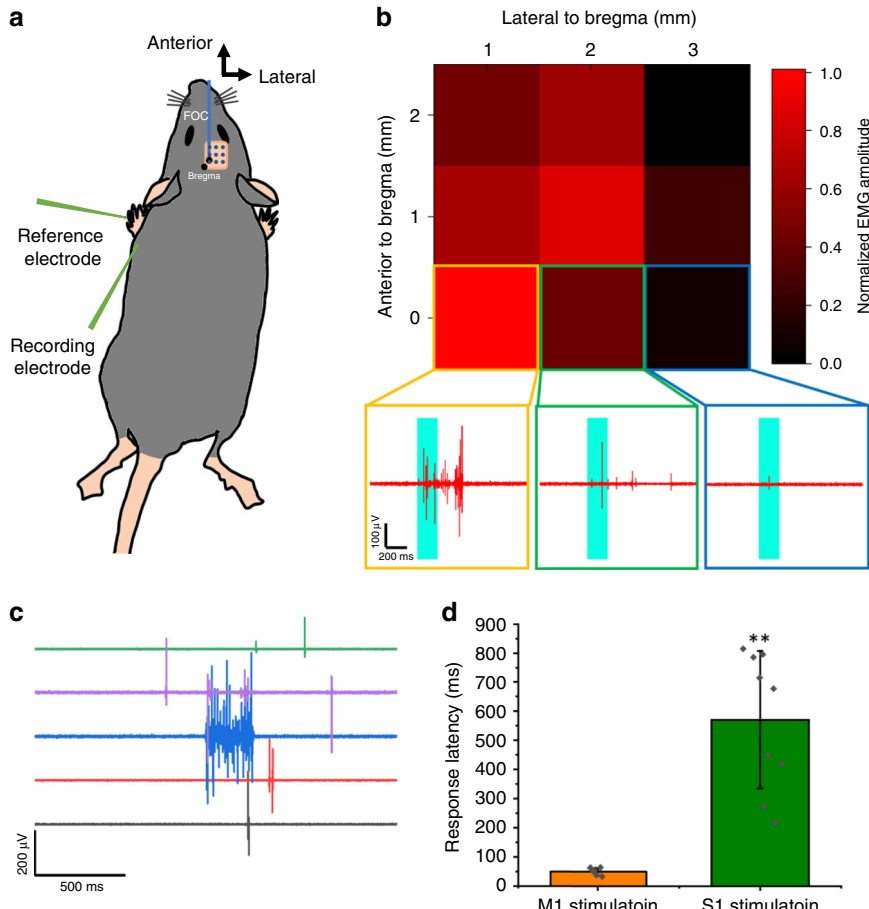

**Fig. 5 Mapping of forelimb representation in the motor cortex by FOC stimulation. a** Schematic of the experiment. **b** Heat map showing maximum peak to peak EMG response amplitude to FOC stimulation on different locations of the motor cortex. Inserts: representative EMG traces from indicated locations. **c** Representative traces of forelimb EMG responses to S1 FOC stimulation. **d** Average delay of EMG responses to M1 and S1 stimulation. Blue box: stimulation duration. Error bars: ± SD. **$p < 0.01$, two-sample $t$-test. Source data are provided as a Source Data file.

**Modulation of motor activity by FOC stimulation.** Since ultrasound is known to modulate motor activities in vivo[5,6,17], we next turned to the motor cortex to investigate the functional outcome of the FOC mediated neural modulation. The FOC was placed on the motor cortex based on stereotaxic coordinates and an electromyography (EMG) electrode was inserted subcutaneously and parallel to the triceps brachii muscle (Fig. 5a). A laser pulse train over a period of 200 ms was delivered to the FOC, while the muscle potential was recorded simultaneously. The FOC stimulation in the motor cortex evoked strong EMG responses ($396.7 \pm 135.8\,\mu$V) recorded from the forelimb triceps brachii muscle, with a mean latency of $49.7 \pm 12.8$ ms ($n = 9$, from three mice). As a control, no responses were recorded without FOC stimulation (Supplementary Fig. 10). To demonstrate the spatial precision of FOC stimulation, we scanned the FOC stimulation site on the motor cortex, to map the motor representation for forelimb triceps brachii muscle. Since the FOC stimulation was confined within a 500 μm radius, 1 mm spacing was chosen between each stimulation location. We scanned a 3 × 3 mm² area, which covers the majority of motor cortex, and obtained the maximum peak to peak amplitude of the EMG response to the stimulation. A heat map of normalized EMG amplitude is shown in Fig. 5b, each point on the map is an average of three trials on the same stimulation location. Inserts show three representative EMG traces of a strong, medium and weak muscle responses recorded at three different locations, respectively. The maximum EMG responses were obtained at

AP (anterior/posterior) 0, ML (medial/lateral) 1, and AP 1, ML 2, which is in agreement with previous motor mapping studies using intracortical micro current stimulation (IMCS), and photo-activation of ChR2 expressing mice[38]. In comparison, we find that FOC stimulation delivered to the S1 cortex also evoked EMG responses on the forelimb muscle, but with a much longer latency ($571.2 \pm 235.9$ ms, $n = 6$, from two mice $p = 1.5 \times 10^{-4}$, two-sample $t$-test) (Fig. 5c, d). The latency was in the same range as studies using IMCS and single cell stimulation in the S1[39]. After stimulation, the brain was extracted for histology. No tissue damage was observed after repeated stimulation (Supplementary Fig. 11). Collectively, these results show that the FOC can differentially modulate the mouse motor cortex and somatosensory cortex at submillimeter spatial precision, which is not currently possible with transcranial ultrasound neural modulation.

## Discussion

We demonstrated a miniaturized FOC that can induce neural activation with high-spatial precision both in vitro and in vivo. The FOC has a ball-shaped coated tip with a diameter of ~600 μm and allows omnidirectional generation of acoustic waves at 1 to 5 MHz and with the acoustic intensity of 0.48 MPa at the fiber surface. Strong spatial confinement of optoacoustic intensity is achieved by the nano-composite diffusion layer and the ball-shaped geometry. The neural response is shown to be neither thermal nor laser-induced. The high-spatial precision of FOC

stimulation was demonstrated by mapping of the representation of the forelimb triceps brachii muscle. With transcranial ultrasound modulation, such mapping was reported to be difficult, if not impossible, in mice due to lack of spatial resolution[5].

An important observation is that the acoustic wave generated by the FOC directly activates targeted cortical areas in vivo, instead of indirect activation through the auditory pathway. This finding is supported by multiple pieces of evidence. First, the FOC is able to stimulate cortical neurons in culture, where no auditory circuits are involved; second, the FOC stimulated targeted cortical area only with <20 ms delay, which is indicative of direct stimulation; third, the stimulation was delivered to the cortex directly, avoiding any possible bone transduction to the cochlear; finally, the FOC stimulation did not induce neural response on the auditory cortex contralateral to the stimulation site, thus eliminating the involvement of the auditory pathway. Our results are consistent with previous studies where focused ultrasound can directly stimulate hippocampal slices, isolated retina in vitro, and activate mouse M1 in vivo with <50 ms delay[5,14,15].

Although the FOC and piezo-based transducer both can generate acoustic waves in the ultrasonic frequency, significant differences exist between these two devices. First, the FOC with a diameter of around 600 μm is significantly smaller than most commercially available ultrasound transducers. The FOC size can be further reduced by using optical fiber with a smaller diameter and reducing the coating layers. The much smaller size allows the FOC to be implantable and can be used for behavior study in live free-running animals, which is impossible with traditional ultrasound transducers. Second, the FOC generates 1-μs pulsed acoustic waves repeated at 3.6 kHz. The duty cycle of this acoustic wave is about 0.36%. This low-duty cycle avoids ultrasound heating of biological tissues, making the FOC device particularly suitable for in vivo applications. However, these two modalities do share some similarities. For most transcranial ultrasound neural modulation applications, the peak pressure ranges from 0.1 to 2 MPa[1,2]. The peak acoustic pressure generated by the FOC was measured to be 0.48 MPa, which falls into the range of ultrasound intensity used for neural modulation reported in the literature. Furthermore, a wide range of frequencies has been reported to achieve neural modulation (200 kHz to 32 MHz)[1,2]. Generally, the lower frequency is used for transcranial stimulation, and a higher frequency is used to achieve high-spatial confinement. The current FOC generates broadband ultrasound wave with multiple peaks ranging from 1 to 5 MHz, where the spatial precision is achieved because the acoustic wave intensity from the FOC tip attenuates rapidly with distance.

The mechanism of optoacoustic neural stimulation is yet to be investigated[1]. Ultrasound neural stimulation and optoacoustic neural stimulation are similar in the way that both methods create a mechanical disturbance on the neuronal membrane and are likely to share the same mechanism. Two mechanisms are proposed for ultrasound neural stimulation: ultrasound-induced intramembrane cavitation[40–42] and activation of mechanosensitive channels[43,44]. Future studies using specific mechanosensitive channels blockers are needed to identify the relative contributions of these two mechanisms.

Finally, we note that the FOC tip is very versatile and can be easily customized for more advanced applications. The propagation of the generated acoustic wave depends largely on the FOC tip geometry. While the ball-shaped FOC tip allows omnidirectional acoustic wave propagation with a rapid attenuation of optoacoustic intensity with distance, other geometries can be adopted to generate forward, focused and even patterned, complex acoustic field[45–47]. These advanced geometries can be used for neural modulation at even higher spatial resolution. Additionally, the fiber-based design allows the FOC to be implanted for longitudinal study in live, behaving animals. Given the increasing popularity of ultrasound neuromodulation, the compactness, cost-effectiveness, and versatility of FOC open a lot of opportunities to utilize the optoacoustic effect to achieve high-precision neural stimulation. Without the need for genetic modification, we expect that FOC will eventually be used for neural modulation on human subjects, similar to electrode-based deep brain stimulation but in a metal-free manner.

## Methods

**Fabrication of FOC.** The FOC was fabricated by first coating the fiber with a light diffusion layer, followed by coating of an absorption layer. ZnO nanoparticles (Sigma-Aldrich) were mixed with epoxy at a concentration of 15% by weight, and a polished multimodal optic fiber with 200 μm core diameter (Thorlabs) was dipped 150 μm into the mixture and quickly pulled out using a one-dimension translational stage under stereoscope. After 30 min curing at room temperature, the diffusion layer (~100 μm) was coated on the fiber tip. The absorption layer was fabricated by dipping of the diffusion layer coated fiber into a graphite powder and epoxy mixture (30% by weight), quickly pulled put and cure in room temperature. This process was repeated to ensure the absorption layer has enough thickness to absorb all the photons, which leads to a final absorption layer thickness of ~200 μm.

**Acoustic wave characterization.** The intensity of FOC-generated acoustic wave was measured with a needle hydrophone with a 0.04 mm sensor size (Precision Acoustics). For mapping of the acoustic wavefront, an EKSPLA OPO Laser with pulse width 5 ns, repetition rate 10 Hz was coupled into the FOC fiber as excitation laser. Photoacoustic signals were acquired in a water tank by a low-frequency transducer array (L7-4, PHILIPS/ATL) and processed by an ultrasound imaging system (Vantage128, Verasonics Inc.).

**Primary neuronal and glial cultures.** Primary cortical neuron cultures were derived from Sprague-Dawley rats. Briefly, cortices were dissected out from embryonic day 18 (E18) rats of either sex and then digested with papain (0.5 mg/mL in Earle's balanced salt solution) (Thermofisher scientific) and plated on poly-D-lysine coated coverslips. For primary neuron cultures, cells were first plated in Dulbecco's Modified Eagle Medium (Thermofisher scientific) containing 10% fetal bovine serum (Thermofisher scientific) and 1% GlutaMAX™ (Thermofisher scientific), which was then replaced 24 h later by a feeding medium (Neurobasal medium supplemented with 2% B-27 (Thermofisher scientific) and 1% GlutaMAX™ (Thermofisher Scientific). Thereafter, the medium was replaced every 3 to 4 days until use. For primary glial cultures, cells were cultured in Dulbecco's Modified Eagle Medium (Thermofisher scientific) containing 10% fetal bovine serum (Thermofisher scientific) and medium was replaced every 3 to 4 days.

**Calcium imaging.** Calcium imaging was performed on a lab-built wide-field fluorescence microscope. The microscope was based on an Olympus IX71 microscope frame, with a 20x air objective (UPLSAPO20X, 0.75 NA, Olympus), illuminated by a 470 nm LED (M470L2, Thorlabs), an emission filter (FBH520-40, Thorlabs), an excitation filter (MF469-35, Thorlabs) and a dichroic mirror (DMLP505R, Thorlabs). Image sequences were acquired with a scientific CMOS camera (Zyla 5.5, Andor) at 20 frames per second. Oregon Green™ 488 BAPTA-1 dextran (OGD-1) (Invitrogen) was dissolved in 20% Pluronic F-127 in dimethyl sulfoxide (DMSO) at a concentration of 1 mM as stock solution. Before imaging, cells were incubated with 2 μM OGD-1 for 30 min, followed by incubation with normal medium for 30 min. During imaging, cells were placed in extracellular solution for cortical neurons containing 150 mM NaCl, 4 mM KCl, 10 mM HEPES, 10 mM glucose, 2 mM CaCl₂ (pH 7.4). For application of BAPTA-AM, BAPTA-AM (Sigma-Aldrich) powder was dissolved in DMSO at concentration of 15 mM for stock solution, and during experiment, the stock solution was added to the extracellular solution to reach 15 μM final concentration before Calcium imaging.

**Animal surgery.** All experimental procedures have complied with all relevant guidelines and ethical regulations for animal testing and research established and approved by the Institutional animal care and use committee of Boston University. Adult (age 14–16 weeks) C57BL/6J mice were used. Mice were initially anesthetized using 5% isoflurane in oxygen and then placed on a standard stereotaxic frame, maintained with 1.5 to 2 % isoflurane. Toe pinch was used to determine the level of anesthesia throughout the experiments and body temperature was maintained with a heating pad. The hair and skin on the dorsal surface targeted brain regions were trimmed. Craniotomies were made on primary somatosensory (S1) (AP −1.34 ML 2.25), primary motor (M1) (AP −0.62 ML 1.5), and primary auditory cortex (A1) (AP −2.46 ML 4.25) based on stereotaxic coordinates using a dental drill and artificial cortical spinal fluid was administrated to immerse the brain. After stimulation and recordings, the mice were perfused with saline and 10% Formalin, and the brain was removed, paraffin embedded, sectioned, and H&E stained for histology.

**Local-field potential (LFP) recording**. LFP was performed using tungsten microelectrodes (0.5 to 1 MΩ; Microprobes). Tungsten microelectrodes were driven to recording sites through cranial windows based on stereotactic coordinates. The electrodes were positioned with a micromanipulator (Siskiyou). Extracellular recordings were acquired using a Multi Clamp 700B amplifier (Molecular Devices), filtered at 0.1 to 100 Hz, and digitized with an Axon DigiData 1550 digitizer (Molecular Devices). For calculation of response latency, the pre-stimulation period in each recording was used to obtain baseline mean and SD. The threshold was determined by mean ± 2 × SD. The latency was determined when the voltage crosses the threshold for the first time.

**Motor mapping and EMG recording**. The FOC was fix to a digital mouse stereotaxic instrument (Stoelting Co.). The bregma was calibrated to be coordinate (0,0), and the FOC was scanned through the mouse motor cortex with 1 mm interval laterally and posteriorly to the bregma. EMG was performed using needle electrode inserted subcutaneously and parallel to the forelimb triceps brachii muscle. Reference electrode was inserted in the footpad. A ground electrode was inserted subcutaneously on the trunk and ipsilateral to the stimulation site. EMG signals were acquired using a Multi Clamp 700B amplifier (Molecular Devices), filtered at 1 to 5000 Hz, and digitized with an Axon DigiData 1550 digitizer (Molecular Devices).

**Data analysis**. Calcium images were analyzed using ImageJ. The fluorescence intensity was measured by selecting the soma. Acoustic waveforms, calcium traces, temperature traces, and electrophysiological traces were analyzed using Origin 2019. All statistical analysis was done using two-sample *t*-test. Data shown are mean ± SD.

**Reporting summary**. Further information on research design is available in the Nature Research Reporting Summary linked to this article.

## Data availability
We declare that the main data supporting the findings of this study are available within the article and its Supplementary Information files. Source data are available in the Source Data file. Extra data are available from the corresponding author upon request.

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

## Acknowledgements

We thank Y. Bai for help with building the fluorescence microscope, S. Cha with the testing on in vivo experiments, and M. O'Connor, M. Hastings for help with neuronal cultures. We also thank the Boston University Experimental Pathology Laboratory Service Core for help with histology experiments. This work was supported by R01 NS109794 to J.C. to X.H.

## Author contributions

J.C. conceived the concept of using the optoacoustic effect for neurostimulation; Y.J. and L.L. designed and fabricated the FOC device; Y.J., H.J.L., and L.L. designed and performed the experiments; Y.J., H.J.L., H.T., C.Y., X.H., and J.C. discussed and analyzed the data; H.M. provided neuron cultures; Y.J. and J.C. wrote the manuscript. All authors discussed and edited the manuscript; X.H. and J.C. supervised the project.

## Competing interests

The authors declare no competing interests.
