## [Peer Review File · Nature Communications]

Reviewers' Comments:

Reviewer #1:

Remarks to the Author:

This is an interesting focused manuscript describing the use of a laser fiber which transduces the light energy into acoustic energy for neuronal stimulation. It allows the method to be highly localized, which could ultimately help sort out some of the effects seen in the transcranial literature. There are a number of edits and clarifications that need to be made.

The last two sentences of the introduction are too strong. Whether the resolution of the ultrasound is "poor" is subjective. Some would say it is very good since it can be done transcranially.

"Fast ultrasound attenuation" doesn't make sense. High or low would be better descriptors of attenuation than fast.

Mega Pa is usually MPa

Page 8, line 157, isoflurane need not be capitalized.

Page 12, line 248, kHz should be used

Page 14, line 291, Precision Acoustics should be capitalized

Page 14, line 292, repetition rate not spelled correctly.

What does a 200 ms FOC stimulation pulse look like? Please add to Fig. 1. Was all the characterization done with the 3 ns pulse? Please characterize with the pulses used for stimulation. Laser pulses of 3 ns were repeated every .3 ms for a total of 200 ms. In the time domain, this is the pulse in 1b convolved with a comb function and multiplied by a rect function. In the frequency domain, this is the spectrum in 1b multiplied by a comb function and convolved with a sinc function. Please provide characterization plots for each stimulus used.

Does the characterization depend on the material that the fiber is immersed in? Is it different for tissue than for water?

What is the estimated pressure or intensity of the ultrasound?

How is BAPTA applied?

Is Fig. S3 supposed to be compared to Fig 2?

How was it determined that the response in Fig S5 was "negigible?"

Why does it appear that there are standing waves in Fig 3? What is the source of standing waves as presumably the FOC does not reflect much.

Can you provide more details on how you localized your stimulation precisely in Fig. 6?

Reviewer #2:

Remarks to the Author:

There has been growing interest in the use of US for non-invasive modulation of neuronal activity and deep brain stimulation. A major technical challenge in this field has been attenuation and

distortion of US by the skull and brain envelopes, degrading the precision of manipulation and focal stimulation, and warranting higher intensities of US, which could damage the tissue.

In this report entitled 'Optoacoustic Brain Stimulation at Submillimetre Spatial Precision' Jiang et al. presents a series of experimental results demonstrating the use of optoacoustic effect for brain stimulation in a murine model. Authors report a new approach which allows superior spatial confinement of the US generated by the optoacoustic effect. Indeed, after the detailed description and characterization of their new device, authors present data on its use in neural culture, in vivo animal brain stimulation and modulation of motor activity with mapping cortical areas. It is concluded that the method can potentially be used for stimulating specific brain regions with unprecedented accuracy, with applications in research and therapy.

General comments

Overall, this is a well-done report, combining a new technology, cleverly and carefully designed experimental studies with data analysis, nicely presented figures, and data-based discussion. The manuscript is well written and figures are convincing. In view of this reviewer, the manuscript presents a substantial advance in the field of brain stimulation and technology development and could be published in Nature Communication, given that the authors address all concerns and points below.

1. Authors do not specify the laser source used for optoacoustic stimulation. This information should be presented. The first sentence of results ... 'composed of a compact 1030-nm, 3-nanosecond laser' is very general (Line 63-65). What is the length of multimodal fiber? Describe also the strategies for miniaturizing further the 600 microns coated tip.
2. References should be updated to include the most recent papers on optoacoustic brain imaging (line 43, 44 and discussion): doi.org/10.1016/j.tibtech.2019.07.012; doi:10.1016/j.celrep.2019.02.020; doi: 10.1016/j.neuron.2017.10.022.
3. Line 84-85 + Fig. 1e authors should clarify the physical grounds for asymmetric US intensity distribution, given that earlier (line 76) the omnidirectional distribution of acoustic waves is mentioned.
4. How the distribution map relates to the size of the FOC tip (geometry and dimensions)? Also, Fig. 1 legend mentions standard deviation (SD) which is not shown. Finally, authors should improve the labeling of the graphs and provide the units where appropriate (e.g. graph d) and p values.
5. Description of statistics is poor and needs to be expanded.
6. Authors should be clear about the terminology used in a different context. For example, figure 2 states that 'optoacoustic waves induced calcium transients', but in the paper calcium transients are induced by acoustic waves only (produced by optoacoustic effects). Or?
7. Figure 4 is too sparse to make an illustration in the main text; authors should consider moving it to the supplement.
8. Line 327-328 states that the brain was removed and sectioned for histology. Authors should show the outcome of these tests in a separate figure (supplement). What kind of histology was done and what has been verified?

Minor points:

1. Line 92 'Days' with small letter
2. Terms such as *in vitro* and *in vivo* always italics
3. Line 206 space before (IMCS); full stop after (Fig. 6 c, d).
4. After - ; - always with small letters (line 228-231).
5. Line 288 - "This process was repeated to ensure the absorption layer has enough thickness to absorb all the photons" - what's (Or would be) the thickness of such a multilayer coating?
6. Line 311: OGD-1 stands for??
7. Lines 308-317 lab built microscope? Components from Thorlabs? What about the objective?
8. Line 348 - version of the 'Origin'?

Reviewer #1 (Remarks to the Author):

This is an interesting focused manuscript describing the use of a laser fiber which transduces the light energy into acoustic energy for neuronal stimulation. It allows the method to be highly localized, which could ultimately help sort out some of the effects seen in the transcranial literature. There are a number of edits and clarifications that need to be made.

Re: We are grateful for the reviewer's positive comments. We have addressed all the concerns by the following clarification.

The last two sentences of the introduction are too strong. Whether the resolution of the ultrasound is "poor" is subjective. Some would say it is very good since it can be done transcranially.

Re: We thank the referee for the comment. We have rephrased the sentence in the main text. For the referee's convenience, we have listed it below:

"...moreover, the presence of the skull will reflect acoustic wave and compromise ultrasound focus, resulting in spatial resolution of a few millimeters, which is insufficient for region-specific brain stimulation in rodents."

"Fast ultrasound attenuation" doesn't make sense. High or low would be better descriptors of attenuation than fast.

Re: We have rephrased fast attenuation to "rapid attenuation of optoacoustic intensity with distance".

Mega Pa is usually MPa

Page 8, line 157, isoflurane need not be capitalized.

Page 12, line 248, kHz should be used

Page 14, line 291, Precision Acoustics should be capitalized

Page 14, line 292, repetition rate not spelled correctly.

Re: We appreciate the referee's comments. We have corrected the typos and the misuse of capitalizations.

What does a 200 ms FOC stimulation pulse look like? Please add to Fig. 1. Was all the characterization done with the 3 ns pulse? Please characterize with the pulses used for stimulation. Laser pulses of 3 ns were repeated every .3 ms for a total of 200 ms. In the time domain, this is the pulse in 1b convolved with a comb function and multiplied by a rect function. In the frequency domain, this is the spectrum in 1b multiplied by a comb function and convolved with a sinc function. Please provide characterization plots for each stimulus used.

Re: Thanks for pointing out the confusion. Yes, the referee is correct. To clarify the 200 ms stimulation, we added a supplementary figure S2 showing the pulse train of light delivered to the fiber. All characterizations in Fig. 1 were done with a single 3 ns pulse.

Figure S4. Illustration of a laser pulse train for 200 ms stimulation.

Does the characterization depend on the material that the fiber is immersed in? Is it different for tissue than for water?

Re: We appreciate the referee's comment. The frequency of the ultrasound will remain the same regardless of the medium, while the attenuation of ultrasound intensity depends on the attenuation coefficient of the medium. All characterizations were done with the FOC and transducer/hydrophone immersed in water. The attenuation coefficient for brain tissue and water are 0.6 dB/cm MHz and 0.0022 dB/cm MHz respectively, which means the ultrasound will attenuate more rapidly with distance, and the acoustic field will be more localized in brain tissues.

What is the estimated pressure or intensity of the ultrasound?

Re: The acoustic pressure was measured to be 0.48 MPa using a needle hydrophone (page 5 middle).

How is BAPTA applied?

Re: Thanks to the referee for pointing this out. We have added a section describing the application of BAPTA in the online methods. For the referee's convenience, we have listed it below:
 "BAPTA-AM powder was dissolved in DMSO at concentration of 15 mM for stock solution, and during experiment, the stock solution was added to the cell culture medium to reach 15 μ M final concentration."

Is Fig. S3 supposed to be compared to Fig 2?

Re: We thank the referee for the comment. Yes, it is intended to compare with Fig. 2b, where the amplitude of calcium response is reduced by addition of BAPTA-AM (Page 6 middle).

How was it determined that the response in Fig S5 was "negligible?"

Re: We thank the referee for raising the concern. We have rephrased the statement to be more accurate, quoted below:

"Addition of 15 μ M intracellular calcium chelator BAPTA-AM significantly reduced the calcium signal (max $\Delta F/F = 2.6 \pm 0.5\%$, $n = 12$) (Fig. S5) compared to neurons without BAPTA-AM ($p = 1.17 \times 10^{-5}$)"

Why does it appear that there are standing waves in Fig 3? What is the source of standing waves as presumably the FOC does not reflect much.

Re: We appreciate the referee's comment. The image is stitching of 6 reconstruction results at different time delays (0.2, 0.6, 1.0, 1.4, 1.8, 2.2 μ s), which gives the acoustic wave propagation distance in water 0.3, 0.9, 1.5, 2.1, 2.7, 3.3 mm. We have clarified it in the manuscript as "Wave fronts at 6 different time delays with 0.4 μ s interval were stitched together in Fig. 3a." and added time stamp in Fig. 3a.

Can you provide more details on how you localized your stimulation precisely in Fig. 6?

Re: We thank the referee for the comment. We have added the following part to the online methods section.

“The FOC was fixed to a digital mouse stereotaxic instruments (51730D, Stoelting Co.). The bregma was calibrated to be coordinate (0,0), and the FOC was scanned through the mouse motor cortex with 1 mm interval laterally and posteriorly to the bregma.”

Reviewer #2 (Remarks to the Author):

There has been growing interest in the use of US for non-invasive modulation of neuronal activity and deep brain stimulation. A major technical challenge in this field has been attenuation and distortion of US by the skull and brain envelopes, degrading the precision of manipulation and focal stimulation, and warranting higher intensities of US, which could damage the tissue.

In this report entitled ‘Optoacoustic Brain Stimulation at Submillimetre Spatial Precision’ Jiang et al. presents a series of experimental results demonstrating the use of optoacoustic effect for brain stimulation in a murine model. Authors report a new approach which allows superior spatial confinement of the US generated by the optoacoustic effect. Indeed, after the detailed description and characterization of their new device, authors present data on its use in neural culture, in vivo animal brain stimulation and modulation of motor activity with mapping cortical areas. It is concluded that the method can potentially be used for stimulating specific brain regions with unprecedented accuracy, with applications in research and therapy.

General comments

Overall, this is a well-done report, combining a new technology, cleverly and carefully designed experimental studies with data analysis, nicely presented figures, and data-based discussion. The manuscript is well written and figures are convincing. In view of this reviewer, the manuscript presents a substantial advance in the field of brain stimulation and technology development and could be published in Nature Communication, given that the authors address all concerns and points below.

RE: We deeply appreciate the referee’s positive comments. We have addressed all the concerns below.

1. Authors do not specify the laser source used for optoacoustic stimulation. This information should be presented. The first sentence of results ... ‘composed of a compact 1030-nm, 3-nanosecond laser’ is very general (Line 63-65). What is the length of multimodal fiber? Describe also the strategies for miniaturizing further the 600 microns coated tip.

Re: We thank the referee for raising the point. The laser has been specified as “a passively Q-switched diode-pumped solid-state laser centered at 1030 nm with the pulse with of 3 ns and pulse energy of 100 μ J (PQSY, RPMC USA).”

The length of the fiber is approximately 0.5 m. The detailed procedure for fabricating the miniaturized fiber tip is described in online methods. Additional details for size control have been added. For the referee’s convenience, we have listed it below:

“a polished multimodal optic fiber with 200 μ m core diameter (Thorlabs) was dipped 150 μ m into the mixture and quickly pulled out using a one-dimension translational stage under stereoscope. After 30 min curing at room temperature, the diffusion layer (~100 μ m) was coated on the fiber tip. The absorption layer was fabricated by dipping of the diffusion layer coated fiber into a graphite powder and epoxy mixture (30% by weight), quickly pulled out and cure in room temperature. This process was

repeated to ensure the absorption layer has enough thickness to absorb all the photons, which leads to a final absorption layer thickness of approximately 200 μm .”

2. References should be updated to include the most recent papers on optoacoustic brain imaging (line 43, 44 and discussion): doi.org/10.1016/j.tibtech.2019.07.012; [doi:10.1016/j.celrep.2019.02.020](https://doi.org/10.1016/j.celrep.2019.02.020); [doi: 10.1016/j.neuron.2017.10.022](https://doi.org/10.1016/j.neuron.2017.10.022).

RE: We appreciate the referee for this suggestion. We have added these works on optoacoustic brain imaging. We weren't able to find doi.org/10.1016/j.tibtech.2019.07.012, so we added another reference: doi.org/10.1038/s41551-019-0372-9 where the authors noninvasively imaged neural activity in rodents using GCaMP as contrast agent, and doi.org/10.1038/s41551-019-0377-4, the latest review on optoacoustic mesoscopy in biomedicine.

3. Line 84-85 + Fig. 1e authors should clarify the physical grounds for asymmetric US intensity distribution, given that earlier (line 76) the omnidirectional distribution of acoustic waves is mentioned.

Re: We appreciate the referee's comments. The omnidirectional distribution was achieved through the diffusion layer. The light intensity distribution after diffusion layer is shown below. Although light was collected at all angles measured, the forward propagating light (0°) intensity was higher compared to other directions, which yielded the ultrasound intensity distribution map in Fig. 1e. To clarify in the manuscript, we added supplementary figure 2 showing the light intensity angular distribution.

Figure S2. Angular distribution of light intensity after diffusion layer.

4. How the distribution map relates to the size of the FOC tip (geometry and dimensions)?

Re: The distribution map of ultrasound intensity is determined by factors including light intensity, absorber distribution, and fiber coating geometry. In this work, methods used to improve the uniformity of ultrasound angular distribution include addition of the diffusion layer, even coating of the absorber, and ball shaped geometry, where the propagation of acoustic wave is perpendicular to the surface. In particular, we found that the US intensity map depends on the density of nanoparticles and we have optimized the nanoparticles concentration to be 15%.

Also, Fig. 1 legend mentions standard deviation (SD) which is not shown. Finally, authors should improve the labeling of the graphs and provide the units where appropriate (e.g. graph d) and p values.

Re: We thank the referee for raising the concerns. The standard deviation in Figure 1e comes from 3 repeated measurements, which gives minimal variation, and small error bars were shown in the graph. Enlarged view shown below. We have updated the units in Fig. 1d and Fig. S1 and included p values where two-sample t-test was used.

5. Description of statistics is poor and needs to be expanded.

Re: We appreciate the referee's advice. We have improved description of statistics to be more accurate, added p values where a two-sample t-test was used.

6. Authors should be clear about the terminology used in a different context. For example, figure 2 states that 'optoacoustic waves induced calcium transients', but in the paper calcium transients are induced by acoustic waves only (produced by optoacoustic effects). Or?

Re: We thank the referee for pointing out this issue. Indeed, all the calcium transients in the paper were all produced by the optoacoustic wave generated by the FOC. We have rephrased "acoustic wave" into "optoacoustic wave" throughout the paper to avoid confusion.

7. Figure 4 is too sparse to make an illustration in the main text; authors should consider moving it to the supplement.

Re: We thank the referee for the suggestions. We have moved Figure 4 to be supplementary Figure S8.

> 8. Line 327-328 states that the brain was removed and sectioned for histology. Authors should show the outcome of these tests in a separate figure (supplement). What kind of histology was done and what has been verified?

Re: We thank the referee for the suggestions. We have added Supplementary Figure 11 showing the histology results, and added a statement in the main text:

"No tissue damage was observed after repeated stimulation (Fig. S11)."

Minor points:

1. Line 92 'Days' with small letter
2. Terms such as in vitro and in vivo always italics
3. Line 206 space before (IMCS); full stop after (Fig. 6 c, d).
4. After - ; - always with small letters (line 228-231).

Re: We appreciate the referee's comments. We have corrected the capitalization and formatting issues raised.

5. Line 288 – “This process was repeated to ensure the absorption layer has enough thickness to absorb all the photons” – what's (Or would be) the thickness of such a multilayer coating?

Re: We thank the referee for the comment. The sentence has been expanded to “This process was repeated to ensure the absorption layer has enough thickness to absorb all the photons, which leads to a final absorption layer thickness of approximately 200 μm .”

6. Line 311: OGD-1 stands for??

Re: We appreciate the referee's comments. OGD is Oregon Green™ 488 BAPTA-1 dextran (page 3 paragraph 2). We added additional definition in the online methods section.

7. Lines 308-317 lab-built microscope? Components from Thorlabs? What about the objective?

We have updated the calcium imaging part in the online methods section. For the referee's convenience, we have listed it below:

“All calcium imaging was performed on a lab-built wide field fluorescence microscope. The microscope was based on an Olympus IX71 microscope frame, with a 20X air objective (UPLSAPO20X, 0.75 NA, Olympus), illuminated by a 470 nm LED (M470L2, Thorlabs), with an emission filter (FBH520-40, Thorlabs), an excitation filter (MF469-35, Thorlabs) and a dichroic mirror (DMLP505R, Thorlabs). Image sequences were acquired with a scientific CMOS camera (Zyla 5.5, Andor) at 20 frames per second”

8. Line 348 – version of the ‘Origin’?

Re: We have updated Origin 2019 in the online methods section.

Reviewers' Comments:

Reviewer #1:

Remarks to the Author:

The manuscript has been revised and is generally quite good and an important contribution to the literature. There are still some points that need to be addressed.

I think the authors were confused by the request of Reviewer #2, point 6. The energy that is emitted from the optoacoustic device is an "acoustic wave." The use of the term "optoacoustic wave" doesn't make any sense. Reviewer 2 is asking for the authors to use the term "acoustic wave," including in Fig 2. Please change back all the uses of "optoacoustic wave" to "acoustic wave."

The responses shown in Fig S7 are about 50% of those shown in Fig S5. It seems that glial cells do show a significant response and that the caption should be changed. It is not negligible.

In Figure 3e, what is the distance and meaning of the second area of activation, the one further from the device?

page 11, line 221, It would be safer to say, "...which is not currently possible..."

Reviewer #2:

Remarks to the Author:

Authors addressed all concerns and made numerous improvements in their revised study. This reviewer is satisfied with the manuscript and does not have any further major comments and recommend the article for publication.

Two minor suggestions:

1. The drawing of the mouse in Fig. 5a is rather amateur; authors may consider replacing with a better one.
2. Adding to the citation list a very recent relevant paper (<https://doi.org/10.1016/j.tibtech.2019.07.012>)

Reviewer #1 (Remarks to the Author):

The manuscript has been revised and is generally quite good and an important contribution to the literature. There are still some points that need to be addressed.

I think the authors were confused by the request of Reviewer #2, point 6. The energy that is emitted from the optoacoustic device is an "acoustic wave." The use of the term "optoacoustic wave" doesn't make any sense. Reviewer 2 is asking for the authors to use the term "acoustic wave," including in Fig 2. Please change back all the uses of "optoacoustic wave" to "acoustic wave."

Re: Thanks to the reviewer for the clarification. We have changed all “optoacoustic wave” to “acoustic wave” in the manuscript.

The responses shown in Fig S7 are about 50% of those shown in Fig S5. It seems that glial cells do show a significant response and that the caption should be changed. It is not negligible.

Re: We have changed the description to “Glial cells show less than 2% responses”

In Figure 3e, what is the distance and meaning of the second area of activation, the one further from the device?

Re: The distance in Figure 3e is the distance from the FOC tip to the cells. We have clarified in the figure legends as “Sorted calcium traces of neurons by the distance from the cell to the FOC.” In Figure 3e, there is only one area of activation. The cells are from the field of view in figure 3c and figure 3d.

page 11, line 221, It would be safer to say, "...which is not currently possible..."

Re: We thank the reviewer for the suggestion. We have modified the manuscript as advised.

Reviewer #2 (Remarks to the Author):

Authors addressed all concerns and made numerous improvements in their revised study. This reviewer is satisfied with the manuscript and does not have any further major comments and recommend the article for publication.

Two minor suggestions:

1. The drawing of the mouse in Fig. 5a is rather amateur; authors may consider respacing with a better one.

Re: we thank the reviewer for the point. We have replaced the figure with a more realistic schematic.

2. Adding to the citation list a very recent relevant paper
(<https://doi.org/10.1016/j.tibtech.2019.07.012>)

Re: we thank the reviewer for the point. We have added the paper as suggested.